# Potential of a Remotely Piloted Aircraft System with Multispectral and Thermal Sensors to Monitor Vineyard Characteristics for Precision Viticulture

**DOI:** 10.3390/plants14010137

**Published:** 2025-01-06

**Authors:** Leeko Lee, Andrew Reynolds, Briann Dorin, Adam Shemrock

**Affiliations:** 1Department of Biological Sciences, Brock University, 1812 Sir Isaac Brock Way, St. Catharines, ON L2S 3A1, Canada; 2AirTech UAV Solutions Inc., Inverary, ON K0H 1X0, Canada

**Keywords:** remotely piloted aircraft system (RPAS), precision viticulture, NDVI, thermal imaging, remote sensing, water stress, canopy size, GLRaV-3, GRBaV, LT50

## Abstract

Grapevines are subjected to many physiological and environmental stresses that influence their vegetative and reproductive growth. Water stress, cold damage, and pathogen attacks are highly relevant stresses in many grape-growing regions. Precision viticulture can be used to determine and manage the spatial variation in grapevine health within a single vineyard block. Newer technologies such as remotely piloted aircraft systems (RPASs) with remote sensing capabilities can enhance the application of precision viticulture. The use of remote sensing for vineyard variation detection has been extensively investigated; however, there is still a dearth of literature regarding its potential for detecting key stresses such as winter hardiness, water status, and virus infection. The main objective of this research is to examine the performance of modern remote sensing technologies to determine if their application can enhance vineyard management by providing evidence-based stress detection. To accomplish the objective, remotely sensed data such as the normalized difference vegetation index (NDVI) and thermal imaging from RPAS flights were measured from six commercial vineyards in Niagara, ON, along with the manual measurement of key viticultural data including vine water stress, cold stress, vine size, and virus titre. This study verified that the NDVI could be a useful metric to detect variation across vineyards for agriculturally important variables including vine size and soil moisture. The red-edge and near-infrared regions of the electromagnetic reflectance spectra could also have a potential application in detecting virus infection in vineyards.

## 1. Introduction

Precision viticulture (PV) refers to the management of the variability within a vineyard, resulting in greater consistency with respect to factors such as yield and quality [1]. PV includes observing the factors responsible for any spatial variation and determining the potential for zoning or managing the variation [1]. Yield and grape quality are influenced by various physiological and environmental factors in the viticulture system such as canopy size, disease pressure, soil type, microclimate, sun exposure, nutrition availability, and water status [2,3,4,5]. Previous research in PV has aimed to identify the spatial variation of these factors in a single vineyard block, such that if the spatial variation can be identified, grapes can be managed differently in these subfield regions to maximize fruit quality and/or vine health or possibly harvest fruit into different lots based on this spatial variation [6,7].

In the environment, natural systems display periodic or structured variation in time and space (i.e., temporal or spatial dependence). The viticulture system is yet another example where spatial patterns can develop [8]. The quality and production of grapes can be significantly affected by the spatial variation of vineyard blocks [9,10,11]. In the Niagara grape-growing region of Ontario, Canada, significant glacial activity has created soils with a high degree of variability [12,13]. The water status of grapevines can be directly impacted by differences in soil hydrology in vineyard soils [14]. A drought can negatively affect the growth of individual vines and their photosynthesis [15,16] while an excessive amount of water can lead to a greater risk of disease [17], poor root growth [18], and leaching nutrients from the soil [19]. Vineyard soil moisture (SM) can differ significantly within a vineyard block [20] and differences in vine water status can be seen throughout the growing season [21]. Different soil types and vine vegetative growth impact the variations seen in vineyard water stress [2,22]. In addition to water stress, other metrics of plant health including chlorophyll levels can be affected by factors such as plant life cycle, air contamination, nutrient status, and pathogens [23]. Differences in vegetation size and structure can be attributed to many factors such as nutrient deficiency [24], water status [25], and disease infection [26].

Another stress facing plants in cool-climate grape-growing regions is cold weather exposure during dormant periods. Grapevines can vary in their winter hardiness, which refers to the cold acclimation process that is triggered by a reduced photoperiod and lower temperatures [27,28]. Winter injury may occur when the temperatures drop below the hardiness level of the vine. Depending on the severity of damage, the vine may be impaired in vegetative and reproductive growth, and suffer increased disease pressure as well as reduced yield [29,30,31]. Understanding the spatial variability of winter hardiness can lead to targeting management strategies to less winter-hardy areas, maximizing the effectiveness and response of the vineyard.

Another important source of vineyard variation is virus infection. Many vineyards around the world suffer economic losses due to virus infections [32]. Grapevine leafroll virus (GLRaV-3) is one such virus that commonly affects grape production in nearly every major grape-growing region [32,33]. According to a study conducted in Ontario, grapevine rupestris stem pitting-associated virus (GRSPaV) was the predominant virus in Ontario vineyard blocks, followed by GLRaV-3, with close to 50% overall infection [34]. There is clear evidence that GLRaV-3 negatively impacts the productivity and grape quality of the infected vines [35,36,37,38]. Grapevine red blotch-associated virus (GRBaV) is another common and detrimental virus in North America [39]. A recent study revealed its detrimental impacts on vine growth and grape quality, resulting in reduced vine size and yield, and reduced sugar, pigment, tannin, and yeast assimilable nitrogen in the berries while increasing berry pH and titratable acidity [40]. In most cases, grapevine viruses are detected in infected leaf tissue by amplifying the viral genome sequence with polymerase chain reaction (PCR) [41]. However, several diagnostic challenges exist with PCR, including that infected vines have uneven symptom distribution, so not all leaves may contain the virus at the same level; new infections are low in virus titre; and multiple viruses can infect grapevines at high rates, requiring multiple tests to detect all viruses present [33,34]. Sampling grapevines and testing is also time consuming and expensive. Since effective control and treatments for these grapevine viruses has not been developed yet, recommendations are to remove infected vines before they act as a source of transmission in the vineyard [42]. Thus, there is an urgent demand to develop an efficient method for the rapid and early detection of viruses.

Conventional methods of detecting vineyard variability involve direct ground surveys by well-trained personnel or specialized equipment. Limitations associated with these methods range from being time consuming and labour intensive, to requiring elaborate field and laboratory procedures and expertise, and differences in measurement accuracies depending on the variable under investigation [24,43]. For these reasons, conventional detection methods of vineyard variability are often not viable for processing large numbers of plant samples due to the time and labour involved in their proper execution [43]. Despite the utility of traditional ground surveys, there remains a high demand for an efficient method for rapid detection of plant stress [44].

A solution to this demand could be the use of remote sensing technologies. Remote sensing technologies are used to measure electromagnetic energy reflected by a surface, such as vegetation or soil, to obtain information about the object [45]. The majority of remote sensing applications involve measuring the reflected radiation compared to the transmitted or absorbed radiation from a plant’s surface [46]; however, plant leaves can also emit energy via fluorescence or expulsion of heat [47]. Photosynthesizing plants require external energy in the form of sunlight to execute their photosynthetic activity. Biological pigments absorb energy from photons of sunlight, which carry energy proportional to their radiation frequency. These pigment molecules are responsible for the colour that we visually perceive from plants, as they interact with specific wavelengths in the visible range of the electromagnetic spectrum, absorbing particular wavelengths for energy harvesting and reflecting all others [48,49,50]. Due to the role of pigment molecules in plant photosynthesis and productivity, pigment content in leaves is a good indicator of plant health and photosynthetic activity [48]. It has been demonstrated that the photosynthetic capacity of a plant is impacted by abiotic stresses such as water stress and nutrient availability [15,48,51], and that the quantities and ratios of pigments in a leaf can be altered in response to changes in photosynthetic capacity [49]. Stresses such as water, nutrients, and viral infection have demonstrated physical symptoms associated with changes in leaf colour and patterns, indicative of changes in their pigment content. These changes would further impact the specific wavelengths of light being absorbed and utilized by the plant, and those being reflected [52,53,54,55,56]. There have been several techniques developed to identify changes in plant health using wavebands composed of EM spectrum data [57]. As an example, the normalized difference vegetation index (NDVI) is based on calculations transferring individual wavelength data into the ratio between near-infrared and red reflectance in each band [58]: NDVI = [(near infrared) − (red)]/[(near infrared) + (red)]. Numerous other remote sensing indices can be calculated from the green, red, red-edge, and NIR regions of EM reflectance to detect vegetative growth and potential vine stress including that due to water, nutrients, and disease [59,60,61,62]. In addition, a thermal sensor is a useful tool in precision viticulture for investigating plant stress levels, as it measures the heat released by plants, primarily to detect water stress as leaf temperature increases when water stress increases [63]. The use of remote sensing thermal imagery in deficit irrigation has also been investigated [64]. Using thermal imagery, it was possible to discern differences in leaf water potential resulting from different irrigation treatments and acquisitions taken under full canopy growth and dry conditions were most effective for monitoring deficit irrigation [65].

Remotely piloted aircraft system (RPAS) flights at low altitude allow the collection of remote sensing imagery at high spatial resolutions, in the range of one centimetre/pixel [66]. Several promising approaches to mapping leaf water potential [25,67] and vegetative growth [68] have been suggested. Vineyard canopy characterization by remote sensing is an improvement over manual characterization of the canopy, ensuring high levels of accuracy, efficiency, and reliability [1,24]. The existing research shows that remote sensing can detect vineyard variability, but there is still a limited literature on its ability to detect certain variables, such as winter hardiness and virus infections. The correlation of remote sensing data to crop quality has been shown to vary across vineyards and vintages; thus, research on different varieties and site-specific methodologies should be investigated [69,70,71,72].

The primary objective of this research was to evaluate remote sensing data including several vegetation indices and thermal data collected from RPAS flights to improve vineyard management. To accomplish this objective, remote sensing data along with measured viticultural data (vine performance and health) were used to determine vineyard spatial/temporal variability and to elucidate relationships between variables.

## 2. Results

### 2.1. Principal Component Analysis (PCA) Between Remote Sensing Data from RPAS Flight and Viticulturally Important Variables

PCA results were derived from the first two factors, which explained between 46 and 67% of the data (Figure 1 and Figure 2). Additional components were excluded from the results to avoid complicating the visualization analysis. NDVI and vine size were clustered together on all six sites in both years. Thermal data were clustered with soil moisture at three sites in 2016. For site 3, NDVI was clustered with vine size and leaf water potential (ψ), whereas thermal data were clustered with soil moisture and LT50 in both years. Other variables were not consistently associated with the remote sensing data in the PCAs. By observing the short vectors for some variables, Pearson’s correlations can be used to determine direct relationships with remote sensing data.

### 2.2. Pearson’s Correlation Between Remote Sensing Data from RPAS Flight and Viticulturally Important Variables

Table 1 shows the correlations between NDVI and viticulturally important variables. Vine size was positively correlated with NDVI in all the sites through both the years. There was an inverse relationship between NDVI and soil moisture (SM) for three sites (site 2, 3, and 5) in both years. However, NDVI was positively correlated with SM at site 4 and there was no statistically significant correlation (95% confidence interval, *p* ≤ 0.05) between NDVI and SM at site 6. There was a positive relationship between remote sensing NDVI and leaf water potential (ψ) at two sites (site 3 and 4) in both years. A positive relationship between leaf ψ and NDVI was also observed at site 2 in 2015. NDVI and leaf ψ did not show any meaningful correlation at site 1 and site 6. There was a positive relationship between remote sensing NDVI and stomatal conductance (Gs) at site 3 throughout both years. NDVI was also positively correlated to Gs at sites 1 and 4 in 2016. However, a negative correlation between NDVI and Gs was observed at site 6 in 2016. There was no statistically significant correlation (95% confidence interval, *p* ≤ 0.05) between NDVI and Gs at site 2 and site 5. There was a negative correlation between remote sensing NDVI and LT50 at site 3 over the years. NDVI was also negatively correlated to LT50 at sites 1 and 2 in 2016. There was no meaningful correlation between NDVI and LT50 at sites 4 and 5.

Table 2 shows the correlations between thermal emission data and viticulturally important variables. There was a negative relationship between thermal emission data and leaf water potential (ψ) at two sites (site 3 and 5) in both years. A negative relationship between thermal emission and Gs was also observed at sites 2 and 4 in only one year. A negative relationship was also seen with vine size in five out of six sites in at least one year per site. However, the relationships between thermal data and the other variables (Gs and LT50) did not show temporal stability or consistent relationships across sites.

### 2.3. Mapping and Spatial Autocorrelation Analysis

Spatially clustered variables may be more appropriate for precision viticulture applications due to the ease of targeting management to larger vineyard clusters/zones rather than sporadically throughout the vineyard [2,73]. The Moran’s Index and associated *p*-values (Table 3) were used to determine spatial autocorrelation for each variable, determining whether a variable was spatially clustered. The Moran’s Index and a *p*-value can be utilized to test if the null hypothesis of randomly dispersed data is true or false at a 95% confidence interval (*p* ≤ 0.05), and the Moran’s I score ranges from −1 to +1, where a positive score indicates clustering and a negative score indicates dispersion.

The NDVI and thermal imaging from remote sensing sites were highly clustered across all six sites. In both years, the SM data were clustered in all six sites, while leaf ψ and Gs were clustered in five sites and not for site 2, where most spatial data were randomly distributed. The data on vine size were highly clustered at all six sites in 2016 but only at three sites in 2015. The winter hardiness data were also clustered at five out of six locations in 2015 and at four sites in 2016. In general, remote sensing and all other variables were highly clustered, with site 2 showing the most randomly distributed pattern.

The data from remote sensing imagery must be interpreted by grape growers, with reliable maps showing areas of substantial variation, in order for the data to be useful in site-specific crop management. Figure 3, Figure 4, Figure 5, Figure 6, Figure 7 and Figure 8 show maps of remote sensing data from RPAS flights and of viticulturally important variables in 2015 and 2016. The NDVI maps for each site were similar across years, showing high temporal stability.

At site 1 (Figure 3), the NDVI maps showed similar spatial patterns to those of vine size in both years with low values in the southwestern zone of the block and high values in the northeastern zone. The NDVI maps showed inverse spatial patterns with LT50 in 2016, where these variables had higher values in the southwestern zone and lower values in the northeastern zone. At site 2 (Figure 4), the NDVI maps showed lower NDVI values in the southwestern and northeastern portions of the block and higher NDVI values in the northwestern zone of the block, and showed highly patchy spatial patterns with high and low values seen throughout the entire site. Similar inverse patterns were seen in maps of soil moisture in both years. The NDVI maps showed similar spatial patterns to those of vine size in both years and of leaf ψ in 2015, whereas the spatial distributions of soil moisture were inversely correlated to those of the NDVI in both years.. At site 3 (Figure 5), the NDVI maps showed similar spatial patterns to those of vine size, leaf ψ, and Gs, with lower values in the north and the centre of the block and higher values in the south. Maps of soil moisture and LT50 showed inverse spatial patterns with respect to the NDVI, with lower values appearing in the south and central-north side, and higher values appearing in the northwest and central-east sides. At site 4 (Figure 6), the NDVI map showed a low NDVI along the northwestern edge of the block and higher NDVI in the southeastern portion of the block. It showed similar spatial patterns to those of leaf ψ, Gs, vine size, and soil moisture in both years. The map of LT50 in 2016 showed an inverse spatial pattern with respect to the NDVI.

At site 5 (Figure 7), the NDVI maps showed a higher NDVI in the eastern side of the block and along the southern edge of the block and lower values in the central and northeastern areas of the block. Similar spatial patterns to the NDVI were seen in maps of vine size while patterns inverse to the NDVI were seen in maps of soil moisture in both years. At site 6 (Figure 8), the NDVI maps showed similar spatial patterns to those of vine size in both years, with low values in the central-east areas and higher values in the west side. The opposite spatial pattern to that of the NDVI was seen with LT50 in 2015. Overall, temporal consistency of the spatial patterns was observed in the maps of remote sensing NDVI in most vineyards even though some sites displayed highly patchy or striped spatial patterns. Other variables like vine size and soil moisture also showed temporal stability in their spatial patterns.

### 2.4. PCR Results of Grapevine Leafroll-Associated Virus (GLRaV)-1,2,3 and Grapevine Red Blotch-Associated Virus (GRBV)

This study also examined the correlation between remote sensing indices from RPAS flights and virus titre. Two GLRaV-3-infected Cabernet Franc vineyard sites were examined. In Appendix A, Table A1, no samples were infected by GLRaV-1 and only one vine was infected by GLRaV-2 across the sites, so all statistical analyses were confined to the GLRaV-3 titre only. This research also examined the correlations between grapevine red blotch-associated virus (GRBV) infection and remote sensing indices from the RPAS flights since the virus became an issue of concern in the region during this study. An end-point PCR test was performed on the three virus-infected blocks. Appendix A, Table A2, indicates presence of GRBV for the three virus-infected sites

### 2.5. Principal Component Analysis (PCA) and Pearson’s Correlation Between Remote Sensing Data from RPAS Flight and Virus Titre

According to Figure 9, over 71% of the data for GLRaV3 and GRBV detection at each site was explained by PCA models based on the first two factors. Three out of five virus-infected sites showed a clustering of the virus titre and NIR/red edge. For the GLRaV-3-infected site 1, the PCA model demonstrated that GLRaV-3 infection was clustered with NIR and red edge but the vectors for GLRaV-3 infection were short, making visual comparisons challenging. For the GLRaV-3-infected site 2, the virus infection was also clustered with NIR, red edge, NDVI, GNDVI, and NDRE while red, green, and thermal showed inverse clustering compared to the virus infection.

At the GRBV-infected site 1, the PCA model demonstrated that GRBV showed an inverse clustering from NIR, red edge, and GRVI while thermal, NDRE, and RTVIcore were clustered with virus infection. GRBV was grouped together with NIR, red edge, and RTVIcore at the GRBV-infected site 2. GRBV presence was also grouped together with green, NIR, red edge, and GRVI at the GRBV-infected site 3. Overall, observing the short vectors for some variables, Pearson’s correlations should be used to test whether other factors can adequately explain the variables.

Table 4 shows the Pearson’s correlation results between virus titre and remote sensing data. Red edge and NIR showed the most correlation with virus titre for both viruses, showing a positive correlation with GLRaV-3 in both sites sampled and with GRBV in two out of the three sites sampled. Interestingly, the first site sampled for GRBV had a negative correlation with these two vegetation indices.

## 3. Discussion

Remote sensing data can identify vineyard variation with temporal stability for certain viticultural variables including water status, virus titre, and canopy size. Even though some sites displayed highly patchy or striped spatial patterns, temporal consistency of spatial patterns was observed in the maps of the remotely sensed NDVI in most of the vineyards, as well as the spatial patterns of vine size and soil moisture, over time.

The relationships between thermal data and many variables (SM, Gs, and LT50) did not show temporal stability or consistent relationships across sites. However, half of the research sites indicated a positive correlation between thermal data and leaf ψ. As previously shown in the literature, thermal imagery can be used to correlate with vine water status since it directly measures canopy temperature, which is related to stomatal closure [74]. Interestingly, four of six sites in 2016, which was a particularly hot and dry growing season, showed inverse correlations between thermal data and vine size, while only two sites in 2015, a year with ample precipitation, showed the same correlation (Table 2). This could demonstrate that higher grapevine water stress is a constraint for vegetation in drought-like growing seasons [2].

The NDVI was most consistently correlated with vine size at all six sites in both years. This is consistent with the literature that reports the efficacy of remote sensing data for detecting vegetative growth such as leaf area coefficient [75], total canopy [76], and the health of vegetation [2]. Plant canopies are largely influenced by the environment and stresses [77]. Thus, grapevine condition and level of stress may be reflected in variations in vegetative growth. Additionally, yield-to-canopy size ratios are often used to diagnose fruit quality indirectly based on balanced pruning [78]. By using NDVI data, grape growers can determine variations in vegetative growth to support them in monitoring vineyard balance [79]. A precise projection of canopy size may allow vine vegetative and reproductive balance to be altered through cluster thinning during the growing season and afterward through pruning tactics [80].

Another interesting observation from this research was an inverse correlation between NDVI and soil moisture (SM). Remote sensing could detect variations in SM at three out of the six sites and showed temporal stability throughout the years. SM values tended to be higher at these sites and there was visual evidence of water accumulation in some spots, mainly caused by heavy clay soil with insufficient drainage. Interestingly, the NDVI was positively correlated with SM in 2015 and 2016 at the site 4 vineyard, located in the Lincoln Lakeshore sub-appellation, with relatively well-drained soil types (glacial till soils with high sand and stone content [81]). There is a possibility of using remote sensing to detect SM for sites where water drainage issues or known high clay content [82] can negatively impact vegetative growth [18,19]. There were fewer sites that showed a consistent correlation between NDVI and leaf ψ; they exhibited a consistent positive correlation at only two sites. With leaf conductance (Gs), the relationship with NDVI was also inconsistent, with three sites displaying positive correlations in 2016, but not showing any stability over time. To examine the feasibility of using remote sensing data for vineyard water management, it is necessary not only to establish a correlation between remote sensing data and vineyard water stress but also to survey vineyards’ soil profiles and their soil drainage capacities. Another variable showing an inconsistent relationship with NDVI was winter hardiness, exhibiting a negative correlation, with temporal stability only at site 3. Three out of the six sites showed significant negative correlations (95% confidence interval, *p* ≤ 0.05) between NDVI and LT50 in 2016, which had a colder dormant season (Appendix A Figure A1), while two vineyards had a correlation in 2015. More winter-hardy vines with low LT50 values were observed at all six vineyard sites in 2016. Water status [83] and pruning weight [84] can influence cold acclimation, but the lack of relationships observed in this research could be explained by the low vineyard variations in LT50 values. There could also be an impact of the dormant season minimum temperature on its detection by remote sensing.

Interestingly, site 3 was the only site where the NDVI was correlated to all measured viticulturally important variables, with these relationships displaying high temporal stability over the two years of study. There were three distinct cultural practices at site 3 that differed from other vineyards. First, the inter-row management at site 3 was topsoil cultivation while the other sites planted cover crops between the rows. The inter-row spectral reflection in the data was not removed for this study, so the NDVI data extracted from the vineyards could have been affected by this reflection [85,86,87]. Secondly, pruning practices and the training system in this vineyard were distinct from other vineyard sites. A canopy-based NDVI measurement can be greatly affected by cultural practices such as pruning and training systems [88]. The spur-pruned cordon systems with stronger horizontal profiles in this site could be more appropriate for remote sensing with an RPAS. There would need to be further research on the effect of training systems on the performance of remote sensing data. Lastly, the application of heavy copper spray was observed through the growing season at site 3. The heavy copper spray created a blue residue on the leaves when the RPAS flight was conducted to obtain the NDVI data. There should be more research on the impact of spray regimes on the performance and timing of remote sensing data collection.

When evaluating the possibility of applying remote sensing data to measure vineyard variability, the development of maps with reliable spatial patterns and potential zonal delineation is essential. In the Moran’s I analysis, viticulturally significant variables and remote sensing data were highly clustered, and zonal vineyard management such as selective shoot thinning or precise spraying based on the clustering across the vineyards may be feasible [1].

The production and quality of grapes are often negatively affected by virus infections in major grape-growing regions. In the absence of effective control or treatment for grapevine viruses, it is urgent to develop a rapid and early detection method, such as using remote sensing data, to detect virus infection. This research confirmed that the remote sensing NIR and red-edge peaks based on RPAS’s multispectral sensor data positively correlated with GLRaV-3 and GRBV infection. The two notable reflectance increases from the virus-symptomatic leaves in the red-edge and NIR spectra can possibly be explained by disorganization of the structure of the mesophyll cell by the accumulation of anthocyanins and carbohydrates from the plant defence response [89]. Internal leaf structure controls reflectance in the red-edge and NIR peaks and the leaf reflectance emerges from the mesophyll cells due to the orientation of the cell walls and the differences in the refractive cell wall and air of the intercellular spaces [90,91]. The cell reorientation results in an increase in the cell wall–air interface and multiple scatterings of radiation and leads to a higher reflectance level in the red-edge and NIR peaks [92,93].

## 4. Materials and Methods

### 4.1. Site Selection

There were six Cabernet Franc vineyard blocks included in this study, located within the Niagara sub-appellations of Ontario, Canada. The sites included Buis vineyard in Four Mile Creek (site 1), Cave Spring vineyard in Beamsville Bench (site 2), Chateau des Charmes vineyard in St. David’s Bench (site 3), George vineyard in Lincoln Lakeshore (site 4), Kocsis vineyard in Lincoln Lakeshore (site 5), and Pond View Vineyard at Four Mile Creek (site 6). Table A3 in Appendix A provides information on differences in vineyard management, block shapes, and sizes across sites. Geolocated sentinel vines (72–81 vines) were mapped on 8 m × 8 m grids within each vineyard by an Invicta 115 GPS receiver (Raven Industries, Sioux Falls, SD, USA). Initially offering 1 to 1.4 m accuracy, this was further improved by post-processing relative to the Port Weller, Ontario, base location, resulting in a precision of 30 to 50 cm.

### 4.2. Electromagnetic Reflectance and Emission Data Collection by Multispectral and Thermal Sensor on RPAS

During veraison in 2015 and 2016, the eBee Classic RPAS (Parrot group, Zug, Switzerland) flew at a maximum speed of 60 km/h with an altitude of 90 m. Data were collected using the Sequoia multispectral and Sequoia thermomap sensors. The multispectral sensor provides four wavebands of green, red, red edge, and near infrared (NIR), with a resolution of 8.47 cm at 90 m altitude. The thermomap sensor measures thermal infrared spectrum emissions at 90 m altitude with a resolution of 0.17 cm. The equipment utilized on the aircraft included a GPS receiver that was attached; a radiation monitor for estimating inbound radiation; and an inertial system designed to maintain the alignment and positioning of images using an autopilot with an operating range of up to 1000 m visually and up to 5 km by radio. The vehicle was powered by an electric motor with a battery life of 50 min. The RPAS and ground control station were provided by Air-Tech Solutions, Inverary, ON, Canada; it performed real-time tracking and image collection. Radiation sensor data, as well as inertial station data, were used for geometric and imaging adjustments such as reflection, image distortions, sun exposure, and vignetting effects. To ensure accuracy and consistency, a geometric correction by ground control points was used to adjust the geometry of the image and adjust the bidirectional reflectance. The inertial station also provided information for the correction of geometric distortions caused by changes in the RPAS attitude and altitude, as well as radiometric correction for the effects of vignetting. Prior to the generation of vegetation indices (VIs), data were adjusted for the input of the sunshine sensor. For each flight, the series of images was assembled into mosaics by selecting the overlapping areas near nadir to keep the viewing angle and directional effects to a minimum. VIs were calculated using the mosaics, and thermal emission data were created in degrees Celsius.

### 4.3. Viticulturally Important Variable Data Acquisition

#### 4.3.1. Soil Moisture (SM)

A time-domain reflectometry (TDR) 300 model from Spectrum Tech. (East Plainfield, IL, USA), in VWC mode, with electrodes of 20 cm length, was used to measure SM at the ground near each sentinel vine. A common method for measuring SM is time-domain reflectometry (TDR) as it is fast, damage-free, and precise in a wide range of soils [94]. With the oscillatory dielectric and electric nature of soil, TDR sends pulses of energy to the soil and measures the return speed, which is negatively correlated with the moisture content in soil. The result is a measure of SM as a proportion of moisture to the overall soil volume. Within a 10 cm radius of each vine, soil samples were taken from both sides and the average values were calculated from three measurements (berry set, lag phase, and veraison) in 2015 and 2016.

#### 4.3.2. Leaf Water Potential (LWP, ψ)

As transpiration occurs and water evaporates from the stomata, there is a build-up of water stress in the xylem because of drought conditions [95,96]. Mid-day LWP is a method to detect vine water stress and was measured with the pressure bomb method, applying pressure to a leaf until droplets of water resurface at the petiole tip [97]. A total of 15–20 vines were sampled within each subset of sentinel vines to establish a consistent grid layout over the sites. To ensure consistency, the readings were taken near solar noon from 10 AM to 2 PM each day when the sun exposure was full capacity [98]. An average of at least three leaves per vine were sampled and leaves were selected from an undamaged primary shoot at the mid-canopy using mature and fully exposed leaves. The sampled leaf was inserted instantly in the chamber of a Model 3015G4 pressure bomb (Santa Barbara, CA, USA) with the tip of the petiole uncovered; a steady increase in pressure (bar) was observed as nitrogen was slowly released into the chamber, recording the pressure measurement as liquid was released from the petiole tip.

#### 4.3.3. Stomatal Conductance (Gs)

Gs was also recorded on 15–20 selected vines using a porometer. This measurement is highly dependent on the plant’s photosynthetic capacity and water status, and relates to sun exposure, turgor and vapor pressure difference, temperature, and atmospheric CO_2_ concentration [99]. A model SC-1 leaf porometer from Decagon Devices Inc. (Pullman, WA, USA) was used to assess Gs in mmol/m^2^ s and it was calibrated with a tool supplied by the producer. Similar to above, three leaves of each vine were selected from an undamaged primary shoot at mid-canopy using mature and fully exposed leaves.

#### 4.3.4. Vine Size

During the winter months, the weights of pruned canes were recorded per vine to determine the vine size [11]. A digital hanging scale was used to systematically weigh each vine’s pruned canes in the field immediately after pruning, providing a weight in kilograms.

#### 4.3.5. Winter Hardiness (LT50)

From each vine, two canes were selected for differential thermal analysis, a technique commonly used to measure the freezing hardiness of a plant’s tissues [100]. The bud LT50 method was used, which identifies the temperature point at which 50% of a plant’s primary buds are killed. Measurements were conducted at three different stages from January to March. Five healthy buds close to the bottom of each collected cane were removed [100]. Each bud was placed on a sample plate and soaked in moist sheets [100]. The plates contained a thermometer to measure the average temperature [101] and each unit included a silicon thermocouple sensor to measure the exothermic spikes which occur when a bud freezes [100]. Afterward, the plates filled with buds were placed in computer-controlled freezers, which started at 4 °C and dropped by 4 °C every hour to −40 °C, and the LT50 was determined for each plate.

#### 4.3.6. Viral Presence for Grapevine Leafroll-Associated Virus (GLRaV)

GLRaV-1, 2, and 3 were tested for viral presence. In September 2016, three leaves were sampled from three different sections of the vine canopy for later virus PCR tests at the Molecular Biology lab at the University of Guelph (Guelph, ON, Canada). The leaves were powdered and placed in liquid nitrogen and frozen at −80 °C. GLRaV-3’s presence was calculated from real-time RT-qPCR results, where a higher score reflects a lower viral presence. In this study, Actin was applied as a reference gene, a reliable marker for grapevines’ response to stress [102].

#### 4.3.7. Viral Presence for Grapevine Red Blotch Virus (GRBV)

The viral presence was measured for GRBV. Two mature canes from each side of the cordon were sampled per vine in December 2018. Viral presence/absence was measured at the virus testing services in the Cool Climate Oenology and Viticulture Institute, Brock University, St. Catharines, ON, Canada, using endpoint PCR. DNA was extracted from composite cane samples using the DNeasy^®^ kit by Qiagen Inc. (Valencia, CA, USA). The samples were PCR-screened using two pairs of GRBV-specific primers: GVGF1 and GVGR1 to amplify a DNA fragment containing the V1 and V2 genes [103],;and GRLaV-4 For and GRLaV-4 Rev to amplify portions of the replicase gene and other genomic segments [104].

### 4.4. Mapping and Data Extraction and Analysis

#### 4.4.1. Mapping

ArcMap 10.6 was used to display the spatial distribution of vegetation indices and thermal images at each site, as well as the IDW interpolation method to create the maps of the sentinel vine point data [105]. In this study, a fixed search radius was used (four sectors with 45° offset; minimum neighbours: 10, maximum neighbours: 15) and the map symbology was classified using quantile breaks to avoid extraordinarily large or empty classes. The variable data were imported and visualized using the World Geodetic System 1984 in ArcMap 10.6.

#### 4.4.2. Remote Sensing Indices Extraction

Multiple remote sensing indices can be calculated based on the green, red-edge, and NIR regions of multispectral imagery to determine whether vines are infected with viruses. In Table 5, feature indices were constructed using the spectral measurements from the RPAS flight to characterize virus detection according to previous studies [59,60].

#### 4.4.3. Global Moran’s I: Spatial Autocorrelation

The spatial distribution of variables at each site was displayed using Esri’s (Redlands, CA, USA) ArcMap 10.6 and Moran’s Global Index, a spatial autocorrelation tool, was used to determine whether the patterns expressed were clustered, dispersed, or random. An analysis of randomly dispersed data is performed via a z-score or a *p*-value, and a Moran’s I score is given from −1 to +1, where a (+) z-score indicates clustering and a (−) z-score indicates dispersion.

#### 4.4.4. Correlation-Based Analyses

A Shapiro–Wilk test was applied to each dataset to verify normality, and any outliers were highlighted on boxplots after evaluation of the data variation. Correlation statistics were generated using XLSTAT v2021 to identify relationships between the remote sensing data and other variables. Pearson’s correlations were computed for all years and site data at 95% confidence levels. Data were standardized before performing principal component analyses (PCAs) to reduce a complex dataset to simple datasets that still describe the original data, where correlated data can be transformed into principal components.

## 5. Conclusions

This study demonstrates that the remotely sensed NDVI from an RPAS could be used to detect the variability within a vineyard for several agriculturally relevant variables. The variable that was most correlated with remotely sensed NDVI, and the most temporally stable, was vine size. The NDVI data can be used to project canopy size accurately and help growers determine an ideal balance of plant vegetation and reproduction. A meaningful inverse correlation with temporal stability between NDVI and soil moisture was also observed for multiple sites. There was evidence at these sites of considerable standing water in areas with inadequate water drainage, indicating that SM can be detected by remote sensing in areas with drainage concerns. Furthermore, only one site showed perfect correlation of the NDVI and all the viticulturally important variables with temporal stability. There were some distinct cultural practices at the site that differed from other vineyards; these may impact the usefulness of remote sensing. Further investigation of the efficacy of remote sensing in PV across vineyard cultural practices will promote accuracy and usefulness of remote sensing data. This research also demonstrated that the red-edge and NIR data had the highest correlation with virus infections, which may indicate their usefulness in confirming virus-infected vines. Future research on these indices for early virus detection with a multispectral sensor should be explored. Overall, the biological relevance of remote sensing data per site and per vintage should still be evaluated through ground truthing of sampled vines. In many cases, it was difficult to use remote sensing data to detect agriculturally significant variables across all sites and vintages.

## Figures and Tables

**Figure 1 plants-14-00137-f001:**
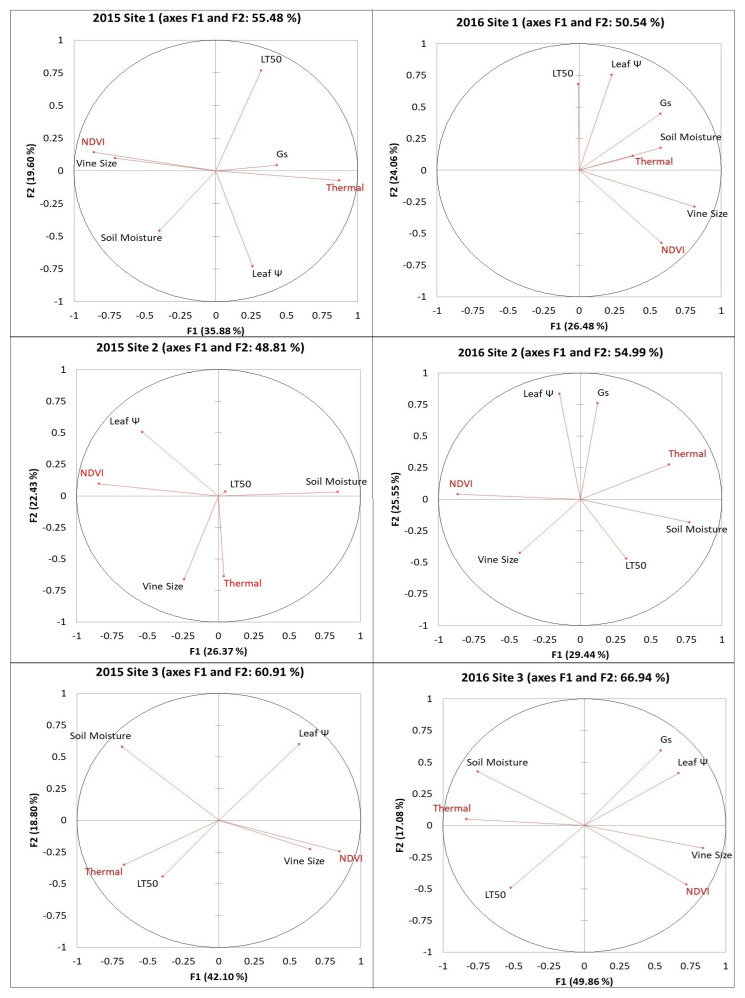
PCA results between remote sensing data from RPAS flight and viticulturally important variables at sites 1, 2, and 3 in 2015 and 2016. Abbreviations: NDVI = Normalized difference vegetation index; Thermal = thermal emission data; Leaf Ψ = leaf water potential; Gs = stomatal conductance; LT50 = temperature that kills 50% of the primary buds.

**Figure 2 plants-14-00137-f002:**
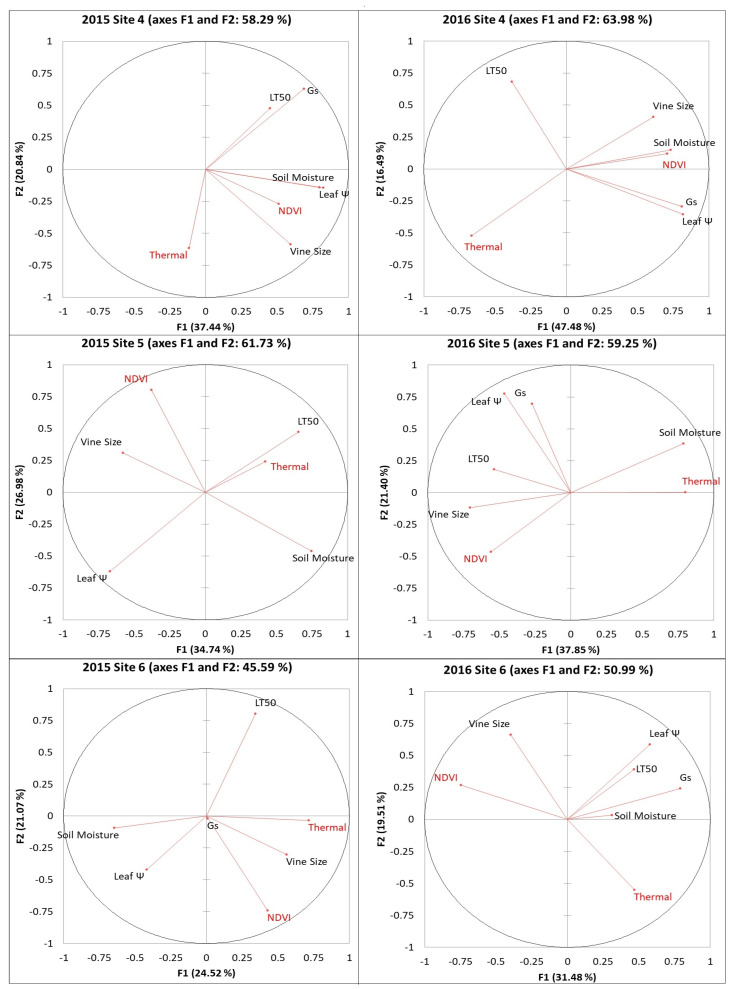
PCA results between remote sensing data from RPAS flight and viticulturally important variables at sites 4, 5, and 6 in 2015 and 2016. Abbreviations: NDVI = Normalized difference vegetation index; Thermal = thermal emission data; Leaf Ψ = leaf water potential; Gs = stomatal conductance; LT50 = temperature that kills 50% of the primary buds.

**Figure 3 plants-14-00137-f003:**
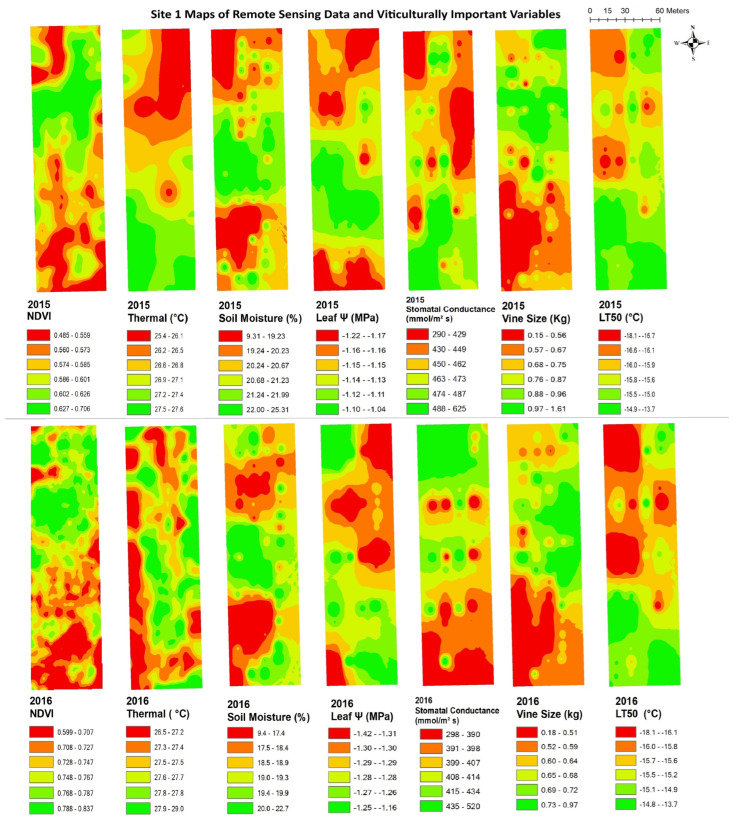
Spatial maps of vineyard variables extracted from remote sensing data and viticulturally important variables at site 1 in 2015 and 2016. Abbreviations: NDVI = Normalized difference vegetation index; Thermal = thermal emission data; Leaf Ψ = leaf water potential; Gs = stomatal conductance; LT50 = temperature that kills 50% of the primary buds.

**Figure 4 plants-14-00137-f004:**
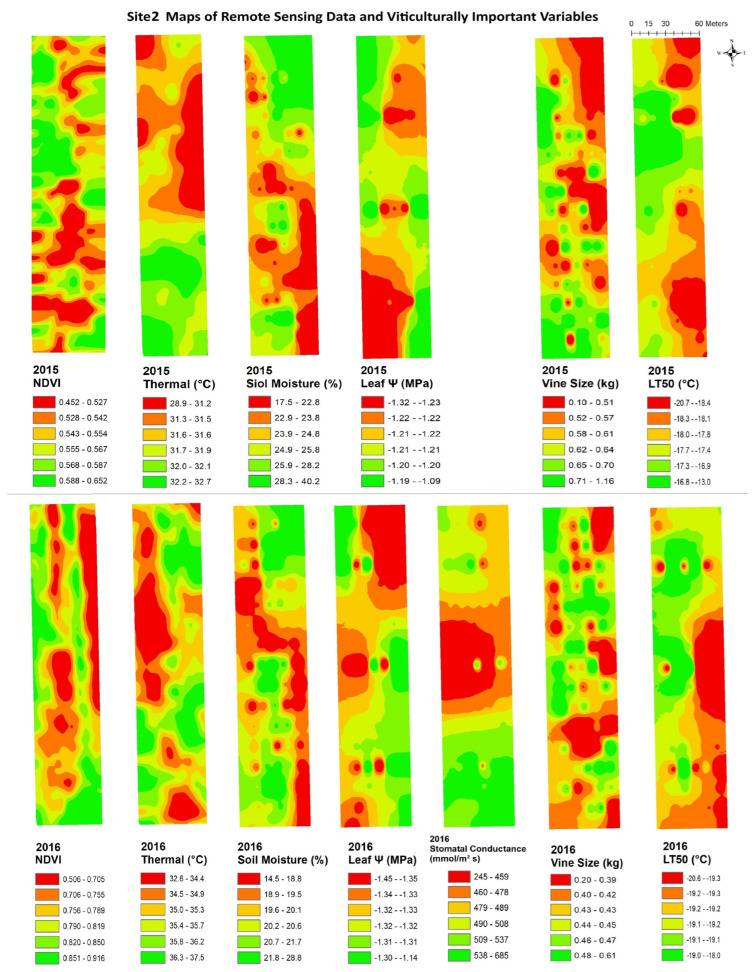
Spatial maps of vineyard variables extracted from remote sensing data and viticulturally important variables at site 2 in 2015 and 2016. Abbreviations: NDVI = normalized difference vegetation index; Thermal = thermal emission data; Leaf Ψ = leaf water potential; Gs = stomatal conductance; LT50 = temperature that kills 50% of the primary buds.

**Figure 5 plants-14-00137-f005:**
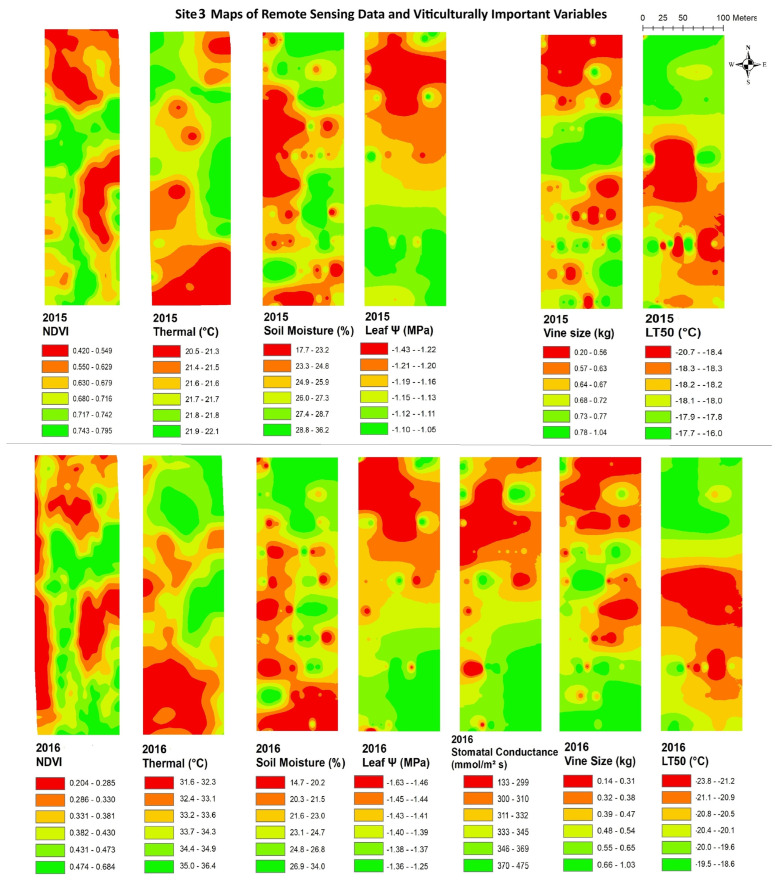
Spatial maps of vineyard variables extracted from remote sensing data and viticulturally important variables at site 3 in 2015 and 2016. Abbreviations: NDVI = normalized difference vegetation index; Thermal = thermal emission data; Leaf Ψ = leaf water potential; Gs = stomatal conductance; LT50 = temperature that kills 50% of the primary buds.

**Figure 6 plants-14-00137-f006:**
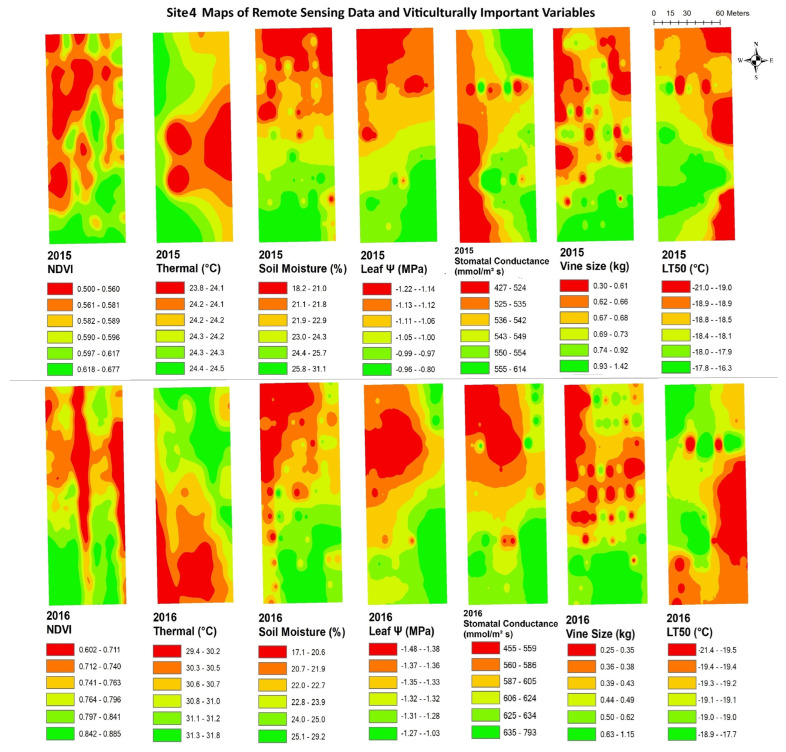
Spatial maps of vineyard variables extracted from remote sensing data and viticulturally important variables at site 4 in 2015 and 2016. Abbreviations: NDVI = normalized difference vegetation index; Thermal = thermal emission data; Leaf Ψ = leaf water potential; Gs = stomatal conductance; LT50 = temperature that kills 50% of the primary buds.

**Figure 7 plants-14-00137-f007:**
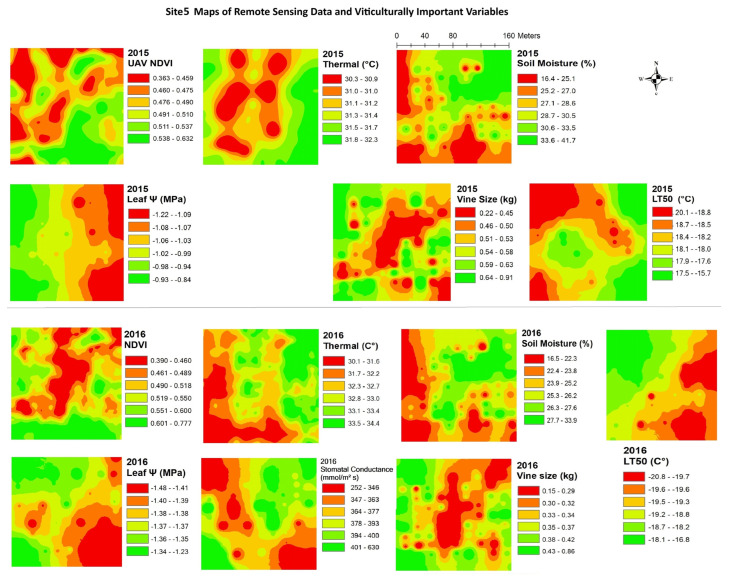
Spatial maps of vineyard variables extracted from remote sensing data and viticulturally important variables at site 5 in 2015 and 2016. Abbreviations: NDVI = Normalized difference vegetation index; Thermal = thermal emission data; Leaf Ψ = leaf water potential; Gs = stomatal conductance; LT50 = temperature that kills 50% of the primary buds.

**Figure 8 plants-14-00137-f008:**
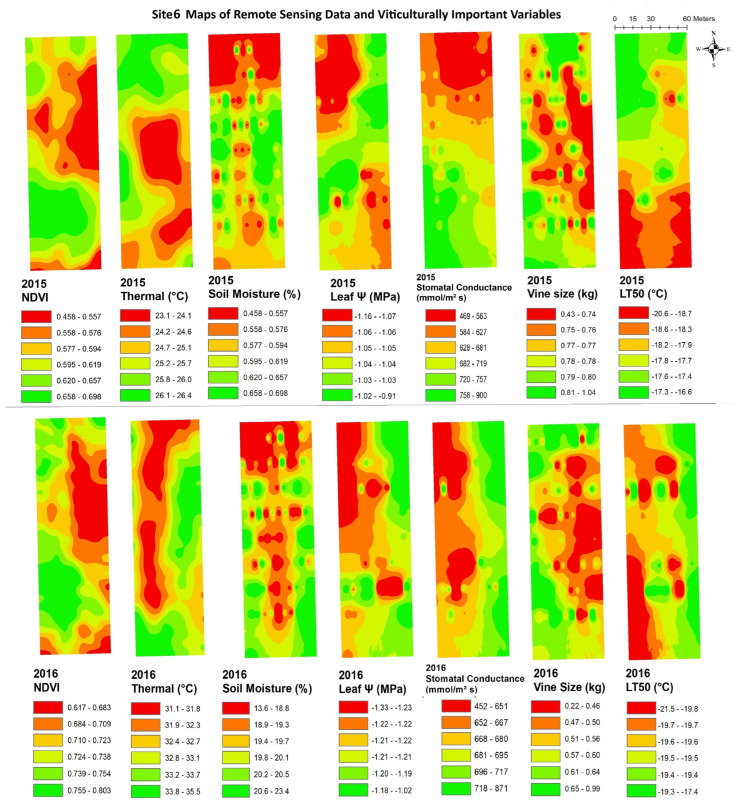
Spatial maps of vineyard variables extracted from remote sensing data and viticulturally important variables at site 6 in 2015 and 2016. Abbreviations: NDVI = normalized difference vegetation index; Thermal = thermal emission data; Leaf Ψ = leaf water potential; Gs = stomatal conductance; LT50 = temperature that kills 50% of the primary buds.

**Figure 9 plants-14-00137-f009:**
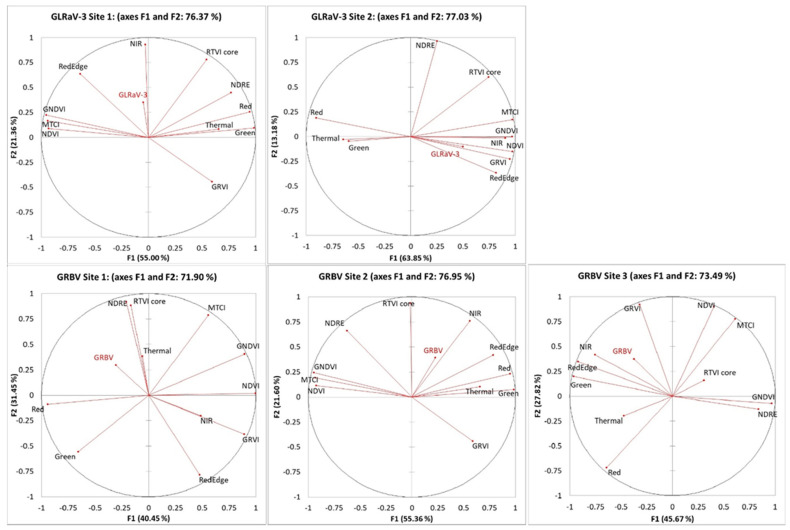
PCA results between virus titres and remote sensing indices in four virus-infected vineyards. Abbreviations: GLRaV-3 = grapevine leafroll-associated virus 3; GRBV = grapevine red blotch-associated virus; NIR = near infrared; Thermal = thermal emission data; NDVI = normalized difference vegetation index; NDRE = red-edge normalized vegetation index; GNDVI = NDVI green; GRVI = green–red vegetation index; MTCI = MERIS terrestrial chlorophyll index; RTVI core = core red-edge triangular vegetation index.

**Table 1 plants-14-00137-t001:** Pearson’s correlation results between remote sensing data (NDVI) and viticulturally important variables in six Niagara vineyards in 2015 and 2016. Those variables with significant correlations (95% confidence) are highlighted in colour, with blue cells representing positive correlation, red cells representing negative correlation, and black cells representing no data collected. Abbreviations: SM = soil moisture; Leaf Ψ = leaf water potential; Gs = stomatal conductance; LT50 = temperature that kills 50% of the primary buds.

	Correlation Matrix					*p*-Value (Pearson)				
Vineyard	SM	Leaf Ψ	Gs	LT50	Vine Size	SM	Leaf Ψ	Gs	LT50	Vine Size
**2015 Site 1**	0.187	−0.165	**0.245**	−0.105	**0.565**	0.106	0.155	**0.005**	0.368	**0.000**
**2016 Site 1**	−0.026	−0.158	**0.300**	**−0.343**	**0.431**	0.823	0.173	**0.009**	**0.002**	**0.000**
**2015 Site 2**	**−0.587**	**0.302**		−0.045	**0.243**	**0.000**	**0.008**		0.700	**0.035**
**2016 Site 2**	**−0.505**	0.053	−0.091	**−0.239**	**0.234**	**0.000**	0.710	0.891	**0.012**	**0.016**
**2015 Site 3**	**−0.661**	**0.351**		**−0.249**	**0.459**	**0.000**	**0.001**		**0.026**	**0.000**
**2016 Site 3**	**−0.705**	**0.501**	**0.376**	**−0.336**	**0.783**	**0.000**	**0.000**	**0.001**	**0.002**	**0.000**
**2015 Site 4**	**0.270**	**0.414**	0.171	−0.038	**0.289**	**0.022**	**0.000**	0.151	0.752	**0.014**
**2016 Site 4**	**0.411**	**0.424**	**0.521**	−0.188	**0.307**	**0.000**	**0.000**	**0.000**	0.114	**0.009**
**2015 Site 5**	**−0.568**	−0.215		0.014	**0.260**	**0.000**	0.052		0.900	**0.019**
**2016 Site 5**	**−0.676**	−0.037	−0.031	0.058	**0.549**	**0.000**	0.745	0.780	0.606	**0.000**
**2015 Site 6**	−0.170	−0.015	**0.279**	**−0.347**	**0.296**	0.133	0.893	**0.012**	**0.002**	**0.008**
**2016 Site 6**	−0.146	−0.063	**−0.293**	−0.109	**0.550**	0.196	0.581	**0.008**	0.338	**0.000**

**Table 2 plants-14-00137-t002:** Pearson’s correlation results between remote sensing data (thermal emission) and viticulturally important variables in six Niagara vineyards in 2015 and 2016. Those variables with significant correlations (95% confidence) are highlighted in colour, with blue cells representing positive correlation, red cells representing negative correlation, and black cells representing no data collected. Abbreviations: SM = soil moisture; Leaf Ψ = leaf water potential; Gs = stomatal conductance; LT50 = temperature that kills 50% of the primary buds.

	Correlation Matrix					*p*-Value (Pearson)				
Vineyard	SM	Leaf Ψ	Gs	LT50	Vine Size	SM	Leaf Ψ	Gs	LT50	Vine Size
**2015 Site 1**	**−0.252**	**0.244**	**0.319**	**0.225**	**−0.433**	**0.028**	**0.034**	0.005	**0.055**	**0.000**
**2016 Site 1**	0.071	0.109	0.043	0.037	**0.329**	0.544	0.349	0.713	0.751	**0.004**
**2015 Site 2**	−0.086	**−0.228**		−0.091	0.129	0.464	**0.049**		0.269	0.063
**2016 Site 2**	**0.274**	−0.052	0.217	−0.152	−0.214	**0.017**	0.655	0.062	0.194	0.065
**2015 Site 3**	**0.270**	**−0.480**		0.155	**−0.266**	**0.016**	**0.000**		0.170	**0.017**
**2016 Site 3**	**0.563**	**−0.469**	**−0.370**	**0.382**	**−0.657**	**0.000**	**0.000**	**0.001**	**0.000**	**0.000**
**2015 Site 4**	0.126	−0.108	**−0.414**	0.086	0.151	0.293	0.369	**0.000**	0.472	0.206
**2016 Site 4**	**−0.390**	**−0.327**	**−0.338**	0.054	**−0.506**	**0.001**	**0.005**	**0.004**	0.654	**0.000**
**2015 Site 5**	0.151	**−0.286**		0.176	−0.041	0.178	**0.010**		0.116	0.719
**2016 Site 5**	**0.661**	**−0.301**	−0.177	**−0.381**	**−0.355**	**0.000**	**0.006**	0.114	**0.000**	**0.001**
**2015 Site 6**	**−0.300**	−0.162	0.023	0.129	0.213	**0.007**	0.150	0.840	0.254	0.058
**2016 Site 6**	0.106	−0.038	**0.270**	0.025	**−0.269**	0.351	0.740	**0.015**	0.827	**0.016**

**Table 3 plants-14-00137-t003:** Moran’s I analysis results (Moran’s Index and *p*-value) for data from remote sensing, yield components, and berry composition in six Niagara vineyards in 2015 and 2016 (95% confidence): blue boxes = clustered, red boxes = random, black boxes = no data collected. Abbreviations: NDVI = Normalized difference vegetation index; Thermal = thermal emission data; Leaf Ψ = leaf water potential; SM = soil moisture; Gs = stomatal conductance; LT50 = temperature that kills 50% of the primary buds.

Moran’s Index							
Site	NDVI	Thermal	SM	Leaf Ψ	Gs	LT50	Vine Size
**2015 Site 1 (n = 76) *p*-value**	**0.6284**	**0.8308**	**0.4617**	**0.6337**	**0.5692**	**0.4670**	**0.3614**
	**0.0001**	**0.0001**	**0.0001**	**0.0001**	**0.0001**	**0.0004**	**0.0004**
**2016 Site 1 (n = 76) *p*-value**	**0.7496**	**0.4990**	**0.4947**	**0.5589**	**0.4372**	**0.2537**	**0.5531**
	**0.0001**	**0.0001**	**0.0001**	**0.0001**	**0.0001**	**0.0089**	**0.0001**
**2015 Site 2 (n = 75) *p*-value**	**0.5505**	**0.6819**	**0.3919**	**0.0318**		**0.4673**	**−0.1343**
	**0.0001**	**0.0001**	**0.0002**	**0.7211**		**0.0001**	**0.3568**
**2016 Site 2 (n = 75) *p*-value**	**0.2315**	**0.6762**	**0.2484**	**−0.0485**	**0.4818**	**0.1978**	**0.2107**
	**0.0225**	**0.0001**	**0.0127**	**0.7858**	**0.0001**	**0.0937**	**0.0395**
**2015 Site 3 (n = 80) *p*-value**	**0.7447**	**0.7829**	**0.5490**	**0.7727**		**0.1782**	**0.4618**
	**0.0001**	**0.0001**	**0.0001**	**0.0001**		**0.1163**	**0.0001**
**2016 Site 3 (n = 80) *p*-value**	**0.8677**	**0.8585**	**0.5660**	**0.6529**	**0.6023**	**0.6718**	**0.6101**
	**0.0001**	**0.0001**	**0.0001**	**0.0001**	**0.0001**	**0.0001**	**0.0001**
**2015 Site 4 (n = 72) *p*-value**	**0.2412**	**0.7988**	**0.6218**	**0.9070**	**0.5269**	**0.5803**	**0.3388**
	**0.0170**	**0.0001**	**0.0001**	**0.0001**	**0.0001**	**0.0001**	**0.0016**
**2016 Site 4 (n = 72) *p*-value**	**0.4626**	**0.7691**	**0.4578**	**0.9176**	**0.6981**	**0.5212**	**0.3591**
	**0.0001**	**0.0001**	**0.0008**	**0.0001**	**0.0001**	**0.0001**	**0.0005**
**2015 Site 5 (n = 81) *p*-value**	**0.3498**	**0.6784**	**0.5167**	**0.8061**		**0.7447**	**0.0803**
	**0.0006**	**0.0001**	**0.0001**	**0.0001**		**0.0001**	**0.3860**
**2016 Site 5 (n = 81) *p*-value**	**0.6561**	**0.7187**	**0.3745**	**0.6455**	**0.4187**	**0.8452**	**0.2290**
	**0.0001**	**0.0001**	**0.0003**	**0.0001**	**0.0002**	**0.0001**	**0.0244**
**2015 Site 6 (n = 80) *p*-value**	**0.6715**	**0.9481**	**0.2923**	**0.6217**	**0.7119**	**0.6743**	**0.1188**
	**0.0001**	**0.0001**	**0.0032**	**0.0001**	**0.0001**	**0.0001**	**0.3129**
**2016 Site 6 (n = 80) *p*-value**	**0.8527**	**0.6785**	**0.3091**	**0.5086**	**0.3138**	**−0.0637**	**0.2336**
	**0.0001**	**0.0001**	**0.0026**	**0.0001**	**0.0015**	**0.6922**	**0.0213**

**Table 4 plants-14-00137-t004:** Pearson’s correlation results between virus titres and remote sensing indices in the virus-infected vineyards. Those variables with significant correlations (95% confidence) are listed in bold, with blank cells representing no correlation, blue cells representing positive correlation between variables, and red representing negative correlation between variables. Abbreviations: GLRaV-3 = grapevine leafroll-associated virus 3; GRBV = grapevine red blotch-associated virus; NIR = near infrared; Thermal = thermal emission data; NDVI = normalized difference vegetation index; NDRE = red-edge normalized vegetation index; GNDVI = NDVI green; GRVI = green–red vegetation index; MTCI = MERIS terrestrial chlorophyll index; RTVI core = core red-edge triangular vegetation index.

Correlation Matrix (Pearson)						*p*-Value				
Index	GLRaV-3		GRBV			GLRaV-3		GRBV		
	Site 1	Site 2	Site 1	Site 2	Site 3	Site 1	Site 2	Site 1	Site 2	Site 3
NDVI	0.102	**0.402**	**−0.244**	−0.097	0.105	0.367	**0.000**	**0.034**	0.390	0.356
Thermal	−0.113	**−0.431**	−0.015	0.217	−0.118	0.316	**0.000**	0.897	0.053	0.298
Green	−0.050	−0.047	−0.036	0.188	**0.375**	0.657	0.698	0.758	0.096	**0.001**
Red	0.013	**−0.312**	0.165	**0.224**	0.020	0.906	**0.008**	0.155	**0.046**	0.864
Red edge	**0.226**	**0.515**	**−0.364**	**0.353**	**0.421**	**0.044**	**0.000**	**0.001**	**0.001**	**0.000**
NIR	**0.254**	**0.531**	**−0.298**	**0.369**	**0.397**	**0.023**	**0.000**	**0.009**	**0.001**	**0.000**
NDRE	0.887	0.045	**0.236**	−0.027	**−0.322**	0.000	0.705	**0.040**	0.815	**0.004**
MTCI	0.431	**0.389**	0.024	−0.091	0.005	0.008	**0.001**	0.836	0.422	0.964
RTVI core	0.366	**0.385**	0.194	0.186	−0.033	0.010	**0.001**	0.094	0.098	0.769
GNDVI	0.173	**0.388**	−0.137	−0.076	**−0.334**	0.024	**0.001**	0.239	0.501	**0.002**
GRVI	0.059	**0.424**	**−0.330**	0.000	**0.349**	0.045	**0.000**	**0.004**	0.999	**0.002**

**Table 5 plants-14-00137-t005:** Remote sensing indices to characterize vine health and viral infections.

Remote Sensing Index	Equation
NDRE (Red-Edge Normalized Difference Vegetation Index)	(NIR − Red Edge)/(NIR + Red Edge)
GNDVI (NDVI Green)	(NIR − Green)/(NIR + Green)
GRVI (Green-Red Vegetation Index)	(Green − Red)/(Green + Red)
MTCI (MERIS Terrestrial Chlorophyll Index)	(NIR − Red Edge)/(Red Edge + Red)
RTVI core (Core Red-Edge Triangular Vegetation Index)	100(NIR − Red Edge) − 10(NIR − Green)

## Data Availability

The data presented in this study are available on request from the corresponding author because the data involves the geo-referenced site information of private landowners.

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
