# Peer review of "Potential of a Remotely Piloted Aircraft System with Multispectral and Thermal Sensors to Monitor Vineyard Characteristics for Precision Viticulture"

_plants, 2025, doi:10.3390/plants14010137_

Round 1
Reviewer 1 Report
Comments and Suggestions for Authors
Dear authors,
the manuscript "Potential of a Remotely Piloted Aircraft System with Multi-spectral Sensors to Monitor Vineyard Characteristics for Precision Viticulture" offers interesting contents regarding the application of precision viticulture to the monitoring and management of the vineyard. I read the manuscript with interest and I reported all my comments and suggestion in the pdf file attached. I would also like to pint out that probably ther is an error in the supplementary matherials which does not appearto belong to this manuscript.

Author Response
I deleted the supplementary table A3.
For other revision, Please see the attachment

Reviewer 2 Report
Comments and Suggestions for Authors
The paper entitled Potential of a Remotely Piloted Aircraft System with Multi-spectral Sensors to Monitor Vineyard Characteristics for Precision Viticulture is thematically focused on the area of ​​precision agriculture with the aim to examine the performance of modern remote sensing technologies to determine if their application can enhance the health and sustainability of vineyards by providing evidence-based stress detection. To accomplish the objective, remotely sensed data such as the normalized difference vegetation index (NDVI) and thermal imaging from RPAS flights were measured from 6 commercial vineyards in Niagara, ON along with the manual measurements of key viticultural data including vine water stress, cold stress, vine size, and virus titer. The objective of the measurements carried out is clearly and concisely defined. The methodological part is presented in a logical sequence and presented in sufficient form. It would be appropriate to add at least basic information about the varieties represented in the evaluated vineyards (only rootstocks are listed in Table 1A). My main comment is on the results and the discussion. It is necessary to make the individual comments more specific and to emphasize the limit values. The graphs in Fig.A1 do not have axis labels, what does the designation historical average mean (what is the length of the evaluated period) mean? The discussion is formulated rather in general terms and would deserve greater precision and confrontation with similar works. I assume that the text and especially the tables and graphs used will undergo editorial editing. In many cases, they exceed the page size, which reduces readability. After incorporating these comments, the text can be accepted for publication.
Author Response
We added statistical limits for the analysis. for example, we added comment on PCA analysis "Considering that the first two components still fairly described the original data, additional components were excluded from the results to avoid complication the visualization analysis."
We also added significant level (95% confidence interval, p<0.05) in the correlation analysis to explain how we decided use the term of "significant". we removed many expressions of 'strong' and 'significant'.
We changed the Figure A1 and added axis labels. We also added comment on the table how the average values were collected. "Historical climate data were derived from the Climate Normals and Average datasets from St. Catharines A station between 1981 and 2010."
We added and sited many evidences from other research to back up our results.
We adjusted the size of the figures to improve readability of the article.
We attached the revised manuscript with the track changes option for your information.

Reviewer 3 Report
Comments and Suggestions for Authors
The article investigates the application of using remotely piloted aircraft
systems equipped with remote sensing devices to assess quality of viticulture.
I do not feel the supplementary document provided is required.
The article is almost backwards in description. A lot of results were
presented, then almost as an after-thought the explanation of the
field equipment and the data processing was produced. The explanation
of the numerical photogrammetric type processing is vague. The biggest
issue is in many places significance is used in describing how the
data relates and almost no description of what statistical limits were
used is defined. Likewise the quality of the data seems very ambiguously
defined. This article has promise it simply needs better ordering
of information and quantification of data quality.
As one sees below the English is spotty in places and as usual as
a reviewer I am trying to read content but English miscues hurt me
interpreting content. So some work on English is required. Some sentences
are long and missing true grammatical corrrectness.
Line 65-68 run on sentence and misses "and" in first section
Line 60 extra space befre accu-
Line 73-74 reverse sentence order
Line 114 vs"."
Line 138 sentence does not have an end
Line 145-147 run on sentence
Line 158-159 state the factors again
Line 158-159 46 and 67% need a statement of quality of those numbers
Line 162 "t"hermal
Line 162 the concept of two years is not well stated earlier
Line 165-166 "is divided by"
Line 166 remove "and precise" as that is subjective
Line 172-187 how did you decide between "sigificant"
Line 193 singular
Figure 1&2 make text larger
Line 232-234 run on sentence
Line 235 "but only at three"
In general - describe how you statistically validated "cluster"
Figure 3&4 - unreadable text
Line 266 "i"nverted
Figure 5&6 unreadable text
Line 281-283 missing some words to make it proper English
Figure 7&8 unreadable text
Line 322 site"s"
Line 338 In "the"
Line 374-377 run on sentence
Line 380-385 run on sentence
Line 397 define statistically what is "strong"
Line 402-404 flip order of sentence parts
Line 409 "survey a map" - strange term
Line 410 define statistically what is weaker
Line 440 define "reliable and precise" in some metric term
Line 471 "by post-processing relative to the Port ...."
Line 478 "at" 90 m"."
Line 476-480 turn this into real sentences
Line 481 monitor (singular)
Line 536 remove 1st "were"
Line 536-537 make 2 sentences
Line 595 define "most"
Line 484-496 Stranger explanation for aerotriangulation of similar
See above. In places the long sentences and sentence syntax made a correct review difficult.
Author Response
Reviewer: I do not feel the supplementary document provided is required.
- Answer: We deleted the supplementary table A3.
Reviewer: The article is almost backwards in description. A lot of results were presented, then almost as an after-thought the explanation of the field equipment and the data processing was produced.
- Answer: We agree with you that this manuscript is almost backwards in description but the Plants journal recommended to put Methods after Results and Discussion.
Reviewer: The explanation of the numerical photogrammetric type processing is vague
- Answer: One of our co-author's association (Air-Tech Solution) did the RPAS data processing part of the research. We can supply the quality report of RPAS remote sensing data processing by Air-Tech Solutions, Inverary, ON as you request. This research was more focused on plant physiology.
Reviewer: The biggest issue is in many places significance is used in describing how the data relates and almost no description of what statistical limits were used is defined. Likewise the quality of the data seems very ambiguously defined. This article has promise it simply needs better ordering
of information and quantification of data quality.
- Answer: We added statistical limits for the analysis. for example, we added comment on PCA analysis "Considering that the first two components still fairly described the original data, additional components were excluded from the results to avoid complication the visualization analysis."
We also added significant level (95% confidence interval, p<0.05) in the correlation analysis to explain how we decided use the term of "significant". We removed many expressions of 'strong' and 'significant'.
Reviewer: As one sees below the English is spotty in places and as usual as a reviewer I am trying to read content but English miscues hurt me interpreting content. So some work on English is required. Some sentences are long and missing true grammatical corrrectness.
- Answer: The English was corrected and many long sentences were simplified.
Thank you for your detailed comments for the correction. We attached the revised manuscript with the track changes option of your recommendation. Please see the review notes with your comments in the revised version.

Round 2
Reviewer 3 Report
Comments and Suggestions for Authors
The critique was dutifully responded to and now deserves publication. The article is still very wordy and some tables are so voluminous no one will read them. An appendix should be after the references